# Health Literacy-Sensitive Counselling on Early Childhood Allergy Prevention: Results of a Qualitative Study on German Midwives’ Perspectives

**DOI:** 10.3390/ijerph19074182

**Published:** 2022-03-31

**Authors:** Julia von Sommoggy, Eva-Maria Grepmeier, Janina Curbach

**Affiliations:** 1Medical Sociology, Department of Epidemiology and Preventive Medicine, University of Regensburg, 93051 Regensburg, Germany; 2Medical Faculty, Institute of Social Medicine & Health Systems Research, Otto von Guericke University Magdeburg, 39120 Magdeburg, Germany; eva-maria.grepmeier@med.ovgu.de; 3Department of Business Studies, Ostbayerische Technische Hochschule Regensburg, 93053 Regensburg, Germany; janina.curbach@oth-regensburg.de

**Keywords:** health literacy, allergy prevention, health professionals, qualitative methods, midwives

## Abstract

In Germany, midwives are involved in extensive antenatal and postnatal care. As health professionals, they can play a key role in strengthening health literacy (HL) of parents on how to prevent chronic allergic diseases in their children. The objective of this study is to explore midwives’ perspectives regarding HL-sensitive counselling in early childhood allergy prevention (ECAP). Twenty-four qualitative semi-structured interviews were conducted with midwives, and data were analyzed using qualitative content analysis. Only a small number of study participants were aware of HL as a concept. However, most of these use screening and counselling strategies which consider individual information needs and which support parental HL. HL sensitivity in counselling is largely based on the midwives’ “gut feelings” and counselling experience, rather than on formal education. The midwives were largely aware of evidence-based ECAP recommendations; however, allergy prevention was not seen as a stand-alone topic but as part of their general counselling on infant feeding and hygiene. They found parents to be more open to receiving complex prevention information during antenatal counselling. In order to strengthen midwives’ roles in HL-sensitive ECAP counselling, their formal education should provide them with explicit HL knowledge and counselling skills. ECAP should be an inherent part of antenatal care.

## 1. Introduction

Health literacy (HL) enables people to make informed health-related decisions to take care of their own health. It is defined as “knowledge, motivation and competencies related to the process of accessing, understanding, appraising and applying health-related information within the healthcare, disease prevention and health promotion setting” [1,2]. A low level of HL is associated with poor health-related knowledge and comprehension, infrequent use of preventive healthcare services, and, especially in older people, poor overall health status and high mortality [3,4]. When it comes to prevention, a higher level of HL is positively associated with health-promoting behavior [5]. It is helpful for patients to understand the connection between health behavior now and health outcomes later, as this encourages them to adhere to prevention recommendations. This is especially challenging when prevention and health outcomes are temporally distant, which is the case for most chronic diseases, e.g., heart conditions or diabetes mellitus. Another example is the prevention of allergies, which will be the focus of this article.

Allergies are chronic diseases which can have a major impact on quality of life [6]. Asthma, eczema, but also allergic rhinitis can present a significant risk to personal well-being [7]. Since the 1990s, an increase in the prevalence of allergic diseases has been observed internationally [8,9]. Allergic diseases are now recognized as a significant public health concern in many developed countries. Research on allergies indicates the importance of early childhood allergy prevention (ECAP), since health-related behaviors in the first three years of life can help prevent allergic conditions [10]. Based on current scientific evidence, exclusive breastfeeding for the first few months, the introduction of solid food after four months, and early exposure to allergens while breastfeeding seem to prevent or lessen the risk of allergies in later life [11]. However, providing advice on allergy prevention is challenging as evidence on risk factors for childhood allergy tends to change quickly due to the high level of research activity in this field [12]. For example, while former guidelines recommended avoidance of allergens (e.g., nuts) and delayed introduction of some solid foods, the current guidelines emphasize that parents do not need to take any specific preventive action regarding their child’s diet [12]. In addition, new myths have emerged from misleading stories in the news and social media, and from product marketers taking advantage of uncertainty [13,14,15].

Although challenging, counselling parents on allergy prevention is important in order to reduce the occurrence of this chronic condition. Health professionals are an important source of information for mothers regarding health-related behaviors [16,17]. Moreover, research has shown that preventive counselling by health professionals can be effective in improving patients’ HL and preventive health behavior, e.g., with respect to smoking cessation or increasing physical activity [18,19,20,21,22]. With regard to the effectiveness of HL-sensitive preventive counselling, a clinical trial conducted by Gharachourlo et al. showed that by counselling in an HL-sensitive way on routine pregnancy care and on modifying lifestyles, healthy lifestyles could be increased in the intervention group [16]. Thus, advice from health professionals may also help prevent chronic allergic conditions [17]. Patients’ HL and health outcomes can be enhanced if health professionals engage in HL-sensitive care, i.e., they take the HL of patients into account when counselling and healthcare settings (private practices, clinical settings) create a shame-free environment for patients with low HL [23,24]. Applying HL counselling techniques, e.g., using simple language and visuals, can help support parents with any level of HL to better understand the information provided [25,26,27]. Thus, HL in general, including that of mothers caring for young children, is not only an individual trait; it is strongly influenced by the healthcare system individuals have to navigate [28].

Enhancing the HL of parents as caregivers is important as research has shown that low parental HL has adverse impacts on child health outcomes [29]. For example, a study by DeWalt et al. on asthma shows that children whose parents had a low level of HL reported more severe asthma symptoms, were more likely to miss school, and were hospitalized more frequently [30]. Another study by Stafford et al. shows a correlation between low HL and maternal intention to exclusively breastfeed [31]. Since breastfeeding is a major contributing factor to the prevention of allergies in later life, mothers’ HL is an important aspect to be addressed in the promotion of allergy prevention [12].

In the German healthcare system, statutory health insurance provides expectant and new mothers with extensive midwife care before, during, and after birth—similar to family nurses in other countries. Midwives provide voluntary antenatal courses advising on childbirth and parenthood, which are offered to pregnant women and their partners. These courses inform participants about breastfeeding and infant care [32] and therefore cover aspects that are also relevant for ECAP [33]. After birth, mothers are entitled to two home visits per day for the first ten days. Thereafter, they can have an additional 16 midwife visits during the first 12 weeks, and eight more up until the end of the ninth month [34]. Data from Bavaria (one of the German federal states this study focuses on) show that 65% (antenatal) and 94.9% (postnatal) of mothers make use of this service [35]. A Czech study showed that pregnant women’s long-term prenatal contact with a midwife was associated with higher HL, suggesting a positive effect of midwives’ interaction with mothers [36]. Due to this very intensive and regular contact with new parents, German midwives are vitally important health professionals when it comes to counselling parents on health issues and the prevention of chronic diseases, especially during this vulnerable phase of transition to becoming a family.

Counselling parents before and after the birth of a child may offer a “window of opportunity” for preventing a whole range a relevant chronic allergic diseases, and midwives could play a key role in strengthening parental HL on allergy prevention [37]. It is thus important to better understand how midwives counsel on ECAP, how they translate current knowledge for parents, and how they ensure that this information can be accessed, understood, appraised, and applied by the parents in their care. To the best of our knowledge, there is, as yet, no research on how German midwives take HL into account in their daily counselling on the health topics relevant to allergy prevention, how they convey scientific evidence, and how they help parents to understand, appraise, and apply this information in their daily life. Hence, the aim of the present study is to gain insight into the experience and practices of German midwives with regard to counselling on allergy prevention and how they support parental HL.

## 2. Methods

An explorative study design was selected in order to capture diverse perspectives and to gain a broad insight into how midwives consider HL in their daily professional practice and how they counsel families on early childhood allergy prevention. Qualitative research allows a flexible approach to the subject, since it enables the researcher to probe interesting facets that come up spontaneously during an interview. Including the personal experiences and subjective views of midwives makes it possible to look at the topic from multiple perspectives. This approach has the potential to capture themes and topics that might not arise when working with predefined, standardized categories and assumptions [38].

We conducted 24 interviews with midwives from May 2020 to March 2021. The initial plan was to conduct face-to-face interviews, but due to the COVID-19 pandemic, which prohibited personal contact, we conducted most of the interviews via telephone. The interviews were semi-structured to enable us to maintain a focus on certain topics, while being open for aspects arising during the conversations. After some minor adaptations and changes following the first three interviews, the same interview guide was used for all interviews to ensure a certain degree of comparability. The interviewees were encouraged to speak freely and discuss their own ideas; thus, the sequence of questions did not always strictly follow the interview guide.

The interview guide (cf. Appendix A) comprised four main topics: 1, information and evidence; 2, knowledge translation and transfer; 3, promotion of health literacy; 3a, counselling of parents and health literacy; 3b, attitudes toward and experiences of health literacy/health literacy-sensitive care; and 4, health literacy concept awareness.

## 3. Recruitment

With qualitative research, the aim is to understand social phenomena in depth, while statistical representativeness is not a requirement [39]. In order to retrieve rich information and ensure rigor, a purposive sampling strategy was used by identifying specific groups of midwives who possessed certain characteristics relevant to the topic being studied, with a view to accessing maximum variation of perspectives (cf. Table 1) [39,40].

First, in urban areas, we expect patients’ backgrounds to be more diverse in terms of socio-economic status, ethnicity, and, since the prevalence of allergic conditions is higher in urban settings, parental information needs are also expected to be different than in rural areas. Thus, the way parents are counselled might differ as well. Bearing this in mind, we intended to recruit from both types of catchment areas. Second, in Germany, there has been a recent shift with regard to the professional education of midwives. Until January 2020, vocational training of three or five years was sufficient to become a midwife. Since then, a bachelor’s degree has become a mandatory requirement. Thus, the education of midwives varies widely, and we assume this might have an impact on counselling and knowledge of health literacy as a concept as well. Our aim was therefore to recruit midwives with different educational backgrounds. Third, we assumed that the length of professional experience would have an effect on how midwives counsel parents and that, in this regard, there might be a significant difference between older midwives with a lot of professional experience and younger, less experienced midwives. Thus, we aimed to include both groups into the sample.

We excluded midwives working solely in clinical settings as the time provided to counsel parents on ECAP in the hospital setting is considered too short. We also excluded midwives with less than two years of professional experience.

To recruit suitable interviewees, contact was established with the Associations of Midwifery in Bavaria, Lower Saxony, and North Rhine-Westphalia. With the support of the associations, a call for participation was sent to the midwives who belonged to these associations and put on the websites. Midwives who contacted us were asked about their education and professional experience, as well as the catchment area of their professional activities and were then, if their answers were appropriate, included in our study. Thus, we had an initial sample of midwives, which was completed by subsequent snowballing and personal contacts. Further cold calling was performed in order to recruit midwives working in rural areas as these were underrepresented in the sample (for further details cf. Figure 1)

After conducting 15 interviews, the research team (JvS, EMG, JC) began to discuss the topics and themes in the data to establish whether data saturation had been reached. The researchers agreed after 20 interviews that no more new topics or themes had emerged. Four additional interviews were conducted to ensure data saturation was reached [41].

## 4. Sample

All of the midwives are female, between 26 and 63 years old, and live in Bavaria (*n* = 17), North Rhine-Westphalia (*n* = 1), or Lower Saxony (*n* = 6). Their professional experience ranges from three to 23 years. Seven midwives in our sample hold a university degree, four have completed special training regarding allergy (prevention), mostly with a focus on breastfeeding. Concerning the catchment areas, 14 interviewees work in villages/small towns (<20,000 inhabitants), ten work in a medium-sized to large town (>20,000 inhabitants) (cf. Table 2).

## 5. Analysis

The duration of the interviews was between 32 and 68 min. A total of 23 interviews were conducted via telephone, and one was conducted face to face. All interviews were audio recorded and transcribed verbatim. Initially, three interviews were jointly discussed and coded by three researchers (JvS, EMG, JC) using ATLAS.ti (v8). Codes were developed deductively, based on the interview guide, and inductively derived from emerging themes in the interview data. After this initial joint coding phase, the rest of the interviews were coded independently by the researcher using the jointly developed codes. Each interview was coded by two researchers. Codes were compared and differences discussed until consensus was reached. This was followed by a thorough and detailed content analysis conducted by the three researchers [42,43,44]. Themes and overarching topics were identified and enriched with the most pertinent quotes [45]. The study follows the COREQ standard for collecting, preparing, and reporting qualitative research results [46].

## 6. Informed Consent and Confidentiality, Ethics Approval

The study has received ethical approval from the Ethics Committee of the University of Regensburg (18-1205-101). All information from the study and informed consent documents issued to study participants were approved by the Ethics Committee of the University of Regensburg. Participation in the study was only possible after providing informed consent to the audio recording and scientific use of the interviews.

## 7. Results

Only five of the midwives interviewed, all of whom held a university degree, were aware of HL as a concept and what it entails. However, although the theoretical concept or the term “health literacy” was not known by many of our interviewees, most of the participating midwives possessed the implicit knowledge and skills needed to identify the HL level and varying information needs of parents.

### 7.1. Assessment of Parental HL and Knowledge on ECAP

In order to explore how and to what extent midwives assess the ECAP-related HL of parents, we asked them to describe how they assess parental HL and how they take differences into account when counselling families on ECAP. Almost all of the midwives reported that it was important for them to somehow assess the pre-existing knowledge of parents regarding the care of their children. Moreover, they took the parents’ ability to absorb new information into account, enabling them to adapt their counselling to the specific needs. The midwives put emphasis on “meeting parents where they are”, which they described as a core skill of midwifery.


*Some want things explained in simple terms, and to others you have to explain or prove everything down to the last comma. I try to convey information in a way that is understandable to them. In other words, I meet them where they are. I don’t need to come up with any scientific explanations for someone who is rather “simple-minded”.*
(Int 3)

Information on parental education and employment is often collected during the anamnesis at the first meeting. Some midwives use this, along with other information, as indicators for parental HL, determining the way information is communicated during their counselling and the level of complexity conveyed.


*Sometimes you just have to be a bit more scientific [with academics] and can give them more facts about sudden infant death syndrome and allergy prevention than with a cashier, where you say, “It’s just good if your child gets breast milk”.*
(Int 12)

None of the midwives used formal strategies to assess parental HL. Most midwives doubt that formal screening would be helpful and are more of the opinion that it could unsettle parents as it might make them feel as though they are being “tested.” The interviewees in our sample preferred to assess parental HL based on their personal “gut feeling”, experience, and intuition.


*This questionnaire stuff… I tend to have the feeling that it would take me away from the women. It would become so scientific. They don’t want that. It’s such a vulnerable time […]. They want to talk a lot. And if you allow that, it’s very easy to understand these women and work with them. You know exactly what you have to say and how you can take good care of them.*
(Int 2)

All interviewees reported that close personal contact with the mothers and families is the most important prerequisite to assess parental HL and information needs regarding ECAP. During conversations and close interaction with parents, the midwives feel confident in their ability to form an accurate impression of parental HL and information needs on ECAP.


*So, you quickly realize in a personal conversation, how much information you can give at once. Whether you can do it en bloc or whether you have to convey it in portions. That is individual, always different. Every person is different. And then you always have to see how it is received. Is it understood? How is it implemented? And then you just have to look at it step by step, how is it working? And that’s how it’s passed on.*
(Int 16)

Besides personal contact, visual impressions can also serve to assess parental information needs and HL. Most midwives perform home visits; thus, the midwives receive a visual impression of the family’s living environment which helps them estimate parental HL and knowledge on ECAP. Specifically, factors such as cleanliness, the child’s sleeping arrangements, visible food and drinks (fruits vs. sweets on the table), accessibility of ashtrays, toys (plastic vs. wood), and pets are considered in the context of assessing ECAP.


*Of course, there is a pattern. First, when I enter, I observe: Is it clean? Is it tidy or chaotic? What kind of furniture does the person have? How is the person dressed? What kind of food, what kind of creams are standing around? These are all things that are subconsciously absorbed. If the baby only wears cotton clothes, then […] you can start on a different level […]. And if, to put it bluntly, you only see plastic toys with a thousand lights and noises and only potato chips and coke, then you have to start off differently.*
(Int 21)

### 7.2. Counselling on ECAP: Evidence-Based Knowledge and Support of Parental HL

Most midwives were aware of having a window of opportunity in early childhood to influence the occurrence of allergies by fostering preventive behavior in parents. Some interviewees were convinced that the timing of counselling on ECAP is crucial for effectively reaching parents. They felt it was most helpful to address new parents as early as possible during the first pregnancy, during antenatal counselling, for example, to provide them with more complex evidence-based (ECAP) information, since parents have more resources to absorb and appraise information before the child is born.


*It is very, very important that I have these conversations [on ECAP] with them during pregnancy, when the woman is full of anticipation and wants to do everything right anyway. Once the child is born and crying and the mother is going completely crazy, it doesn’t really help anymore. It’s very important to find a good time during pregnancy when the woman is eager to learn and the partner is hopefully also there. Then you can reach them really well.*
(Int 15)


*I think you actually only have a chance to really reach a first-time mother. A second or third-time mother feeds as she sees fit anyway.*
(Int 21)

Concerning the midwives’ roles as professional health counsellors, the interviewees saw themselves as providers of scientifically sound information on ECAP and other health-related subjects and as such encouraged the parents to trust their advice.


*I always say, “I explain everything. If there’s anything you don’t understand, please ask.” I also always tell parents, “Better to ask me before you Google anything on the Internet. I’m the expert. I can answer that for you and I’ll try to answer it in a way that you understand”.*
(Int 10) 

The midwives in our sample were generally aware that allergies are a chronic disease that can potentially have a significant negative impact on someone’s quality of life. There are differences in perceptions of what actually constitutes an allergy though. Some midwives did not consider hay fever an allergy, which became clear when they were asked if they themselves had allergies.

Apart from this, almost all midwives seemed to be aware of current recommendations regarding allergy prevention and knew about the shift in evidence regarding exposure to allergens. However, they reported rarely counselling on ECAP directly. When we asked the midwives whether and how they helped parents to access information on ECAP, most midwives reported that ECAP was not a stand-alone topic for them but was included in the counselling on health behavior in general. The most important issues in the context of allergy prevention are nutrition, especially breastfeeding or choice of formula, the (early) introduction of solid food, hygiene, use of cosmetic products (e.g., cream, detergent, wet wipes), and the avoidance of smoking.


*There are a lot of allergic people. And of course, all parents want the best for their child. It [ECAP] definitely comes up in conversations, mostly when talking about breastfeeding versus bottle feeding. If formula is used, the question is what kind of formula is considered best and then it [ECAP] comes up again when solid food is introduced.*
(Int 22)


*Allergy counselling plays a very, very minor role, mostly with people who are already allergic themselves. That’s been the case since this allergy guideline changed. I think that the behavior in the case of allergy risk or higher allergy risk is actually identical to the recommendations for people with a low allergy risk, so it’s not such a big issue.*
(Int 19)

The midwives believed that parents found ECAP-related topics easy to understand. Most midwives assumed that the fact that breastfeeding is the best nutrition for children is common knowledge and does not have to be explained scientifically. Their counselling focuses more on how to establish a comfortable breastfeeding situation for mother and child to ensure exclusive breastfeeding for at least four months, without providing any scientific information on why. Similarly, for the introduction of solid foods, the midwives focus their counselling mostly on how to start, what to begin with, and how to continue, and less on the reasons why.


*It [ECAP] automatically resonates in our work, because we advise on breastfeeding, for example. That is God-given allergy prevention. That’s why it’s always part of our work. And again, when we advise on weaning, it comes quite automatically that you say, okay, start feeding different solid foods in a short time while maintaining the protection of breast milk.*
(Int 2)

When explicit information on ECAP is provided, most midwives have to help parents appraise this information. Most midwives reported having the impression that they needed to put official guidelines and recommendations into perspective, to encourage parents to see them as a blueprint and not as a rule that needs to be strictly adhered to. For example, even if parents understand that early introduction of solid food might help prevent allergies, the child’s readiness to actually eat solids may put constraints on the officially recommended blueprint.


*I try to avoid these blueprints a bit, because I find that most children simply don’t eat according to a blueprint. And then the whole text on allergy prevention doesn’t help if this child decides it doesn’t like vegetables.*
(Int 9)

Thus, the key message most midwives want to give to parents regarding specific ECAP counselling is to “calm down” and rely on their parental intuition instead of trying to strictly follow evidence-based recommendations. They try to help parents appraise information and thus, make the right choices for themselves and their children.


*I think the most important thing is that they don’t get carried away. Instead, they should take a more nuanced look at what information is available and what is really true. And I try to instill a bit of calmness in the parents when it comes to allergy prevention.*
(Int 14)


*I think it would be much more important to strengthen the women in their skills and abilities, for birth or raising children, because the women simply try too hard to follow blueprints and rules, because they read in a self-help book or because it was done like that in the past.*
(Int 9)

Most midwives see themselves as advisers, but they prefer to leave the final say in health-related decisions to the parents, at least as long as the decisions are not potentially life threatening for the babies.


*If a woman decides against breastfeeding, then it doesn’t help if I say, the probability of allergies occurring is much higher. This doesn’t help at all. You can’t convert people.*
(Int 5)


*I emphasize the arguments again, why she should not drink coke. And then I leave it at that. I leave the responsibility to the woman. It is her life, her child, her decision.*
(Int 20)

Sometimes midwives seek compromises, but rather than taking an exclusively “top-down” approach to educating parents on what is evidence based, they aim at agreeing with parents on pragmatic and actionable recommendations, which take the family situation as well as individual preferences of the child and parents into account.


*And also, when women say they don’t want to breastfeed, I say, “That’s not a problem at all. That’s your decision. But it would be great if we could put the baby to the breast at least once in the delivery room.” And no mommy has turned me down yet.*
(Int 21)

### 7.3. HL-Sensitive Counselling Techniques and Materials

Health literacy-sensitive counselling techniques can help support parents in accessing, understanding, appraising, and applying information on ECAP, but also regarding health behavior in general. In the interviews it became clear that, even though most midwives are not aware of HL-sensitive counselling techniques as such, HL-supportive strategies are sometimes applied, e.g., using plain language, omission of scientific wording, etc. These interviewees also reported using these techniques not only when talking to parents with lower levels of education, but in general, since, in their experience, parents with newborn children sometimes have difficulties processing information in this challenging phase of life.


*I really try to avoid medical terms and speak in simple language. […] When I’m in an academic family, it’s sometimes quicker for me to just blurt out the medical terms. But even then, if you are a parent yourself, you sometimes can’t think straight. No matter what you do for a living. And then it’s important that the person advising you simply takes a moment and sits down again and explains it simply, so that everyone understands it.*
(Int 18)

Most midwives described the relationship with the families as being based on mutual trust. Due to the close and intimate contact, they assume that most women and families feel comfortable to ask any question they are concerned with. Hence, the midwives do not see the need for special techniques to encourage parents to ask questions.


*I don’t have the impression that parents have any inhibitions about asking me anything.*
(Int 10)

Only a few midwives knew about and had tried to apply specific HL-sensitive counselling techniques, such as the teach-back method. However, they emphasized that they had done this subliminally, without the parents noticing.


*But then I say, “We’ll make a plan for breastfeeding. Now we have just discussed it, and now I would like to know how you would do it?” In other words, you don’t ask directly, but you do it in a hidden way.*
(Int 15)

Some midwives disliked the idea of using teach-back because it appeared too much like an exam situation to them and they feared it would be rejected by the women in their care.


*You know whether they understood or not. I’m not testing them. I don’t want to behave toward the women like a schoolteacher.*
(Int 11)

Using visualizations is reported as being helpful in supporting explanations, especially with non-native speakers of German, but ECAP-specific pictograms or similar materials are difficult to obtain. Apart from demonstrating things using models or pictures, some interviewees emphasized that actually performing a task with the parents, or supervising and guiding them, has the biggest effect on parental learning.


*Now, for example, when it comes to breastfeeding, I have various materials with me. For example, I have little balls that represent the stomach. So I always try to make it very vivid.*
(Int 15)


*A good example is always when the children are crying and I take them in my arms and speak to them in a calm voice. Then you simply practice this with the parents.*
(Int 8)

Some midwives felt that written material was valuable as a way of providing parents with information, as, due to tiredness, mothers are sometimes not capable of really absorbing the information when the midwife is present. Written information material can be read in more appropriate moments and thus might be better understood.


*I have worksheets and I leave those with the mothers, because I know, when you don’t get enough sleep, all you hear is “bla bla” and you can’t remember what you were told.*
(Int 2)

However, most midwives doubted the benefit of handing out written information, as new mothers and families are often overwhelmed by the high amount of information they receive concerning their children.


*The women get a lot of information material. I have been working in postnatal care for many years and I know that all this paper stuff is just lying around at home. During the period immediately after birth women are so preoccupied with themselves and the baby that there is very little time to look at brochures, etc.*
(Int 10)

In conclusion, the interviews revealed that HL-sensitive counselling techniques are applied to some extent, but not in a systematic or deliberate way.

## 8. Discussion

In the interviews it became clear that most midwives in our sample are not aware of HL as a concept with regard to ECAP. However, the interviewees reported that due to the comprehensive in-home care provided by midwives in Germany before, during, and after birth, their relationship with the parents is often close, thus counselling can be adapted to individual needs. Working with different types of parents was seen by the study participants as a core task of midwifery. However, assessing parental HL and counselling in an HL-sensitive and HL-supportive way is not performed systematically. Personal interaction seems to be the most important factor in establishing a basis for tailor-made counselling on ECAP, frequently supplemented by midwives’ impressions of social and educational backgrounds as well as their visual impressions of the living environment.

The study also shows that midwives are aware of the possibility and importance of allergy prevention during the first months of life. They are mostly acquainted with current recommendations regarding ECAP, however, allergy prevention itself is not treated as a stand-alone topic but is mostly covered in counselling on other topics regarding infant care. Midwives consider it their duty to convey scientific information to parents and help them access, understand, appraise, and apply information, but they view their own influence on the prevention of chronic diseases as limited. They try to find compromises between official scientific recommendations and parental wishes and possibilities. Timing is seen as an important factor when it comes to providing effective advice on allergy prevention, with this preferably being done early on during the first pregnancy.

Most midwives are unfamiliar with HL-sensitive counselling techniques; however, some techniques are applied based on counselling experience and “gut feeling”. Written information is perceived ambiguously.

A lack of awareness of HL as a concept among health professionals was also found in a systematic review by Rajah et al. on the perspectives of healthcare providers and patients on health literacy: The majority of the 19 studies included in the review reported inadequate knowledge and understanding of HL among health professionals. Rajah et al. also found several studies highlighting that health professionals do not regularly use formal HL assessment tools in their practice, but do use other assessments such as verbal cues, nonverbal cues, and their “gut feelings”, which is similar to what was described by the midwives in our sample. The authors of the review also conclude that training of health professionals on HL and HL counselling techniques could help support patients’ HL. The results of our study concerning the assessment of parental HL based on “gut feeling” and work experience are also in line with the findings of an Australian survey on how midwives assess maternal HL [47]: Out of 307 study participants, the majority (77.1%, *n* = 221) reported paying limited attention to formally assessing women’s health literacy.

In our interviews it became clear that especially when it comes to topics that are assumed to be “easy” to understand, such as breastfeeding or the introduction of solid foods, HL is not something health professionals focus on. This lack of attention devoted to HL might lead to a systematic overestimation of parental HL and thus misunderstandings and, as a result, a lack of knowledge among parents, which is in line with the results of various studies on health professionals [48,49,50,51]. For example, Dickens et al. showed that, without using a validated HL screening tool, nurses tend to overestimate patients’ HL. Using the specific measurement tool “Newest Vital Sign” based on a nutrition label and six questions about it, the study showed that 63% of the patients included in the study had a high likelihood of limited HL, whereas nurses only identified 19% as such [49].

The assumption of midwives in our sample that parents are confident enough to ask any questions they have might in some cases also be misleading. Katz et al. showed in their mixed-methods study on patient-physician encounters in a hospital, that patients with a low HL level tend to ask fewer questions and thus might be less informed [52]. The relationship between midwives and families during in-home ante- and postnatal care in our study differs significantly from clinical encounters. That said, the midwives’ “gut feelings” might still convey a false impression regarding parents’ behavior and their ability to ask questions.

Wilmore et al. drew the same conclusions, but with regard to written material. Other than in Germany, Australian midwives seem to rely more strongly on written information material, which is distributed in the 8–12th week of gestation [53]. The Australian midwives who participated in this study were aware that this information needed to be tailored to the individual parents’ needs, or at least needed to be explained according to the HL level of the parents. Similar to our study, Wilmore et al. observed that there was no specific HL screening applied and that midwives were often unaware of their patients’ HL and thus not always able to provide enough support to ensure understanding [53].

The midwives in our study see themselves as responsible for supporting parents in preventing chronic health conditions in their children later in life. They also emphasize that they see the time of transition to parenthood, especially during pregnancy, as a window of opportunity for effective health counselling. In line with this, Phelan et al. describe pregnancy as a point of transition in life, which may be a “teachable moment” and as such an opportunity to positively influence health behavior [54]. During a “teachable moment” individuals can be motivated to spontaneously adopt risk-reducing behaviors. It can facilitate promoting a healthier lifestyle, e.g., healthier nutrition behavior to prevent excessive weight gain during pregnancy, but also preventive behavior, e.g., regarding allergy prevention, as women are open to learning about health-related topics during this transitional time of becoming a parent [54].

## 9. Practical Implications

Midwives might have the opportunity to strengthen families’ HL and thus health-related behavior aimed at the prevention of chronic diseases like allergies. In order to enable them to perform HL-sensitive counselling more systematically and effectively, formal education of midwives on HL as a concept and on HL-sensitive counselling techniques would be beneficial.

It is important to convey to them the importance of systematically assessing parental HL to prevent overestimation. Thus, adequate screening instruments or strategies need to be identified and included in the formal education of midwives. Additionally, HL-sensitive counselling techniques should be explained to midwives, and they should be given the opportunity to practice these during their training in order to enable them to adequately counsel all families. Midwives with less than two years of work experience were not included in our study. However, we believe that integrating HL and HL-sensitive counselling into the curriculum of midwifery training would be especially helpful to young health professionals at the start of their careers, because these cannot draw on work experience when counselling parents with different HL levels.

Concerning ECAP, a useful approach would be to provide German midwives with tailor-made, easy-to-access evidence on ECAP and to integrate ECAP as a stand-alone topic in antenatal counselling.

All of these educational measures could strengthen the role of midwives in Germany in preventing chronic diseases by using the window of opportunity in ante- and post-natal care for effective, HL-sensitive preventive counselling.

## 10. Strengths and Limitations

To our knowledge, this is the first qualitative study with the aim of understanding how German midwives engage in preventive counselling, how they take HL into account, and how they apply HL counselling techniques. It is the nature of qualitative research to draw on a small sample of participants. Therefore, our results cannot be generalized for all German midwives. However, qualitative studies do not claim to produce representative data but are meant to provide an in-depth insight into a specific topic. Our interviews, lasting up to one hour, were very much in depth and enabled us to gain a thorough understanding of the daily working life of midwives. Additionally, we were able to recruit a broad sample of midwives regarding age and experience and could thus capture a wide variety of different perspectives. Moreover, we were able to recruit midwives from different regions of Germany. However, it is not representative for all midwives. We supplemented our initial recruiting via the German Associations of Midwifery in a specific manner, by contacting midwives individually via cold-calling and personal contacts, while considering the criteria for inclusion.

Due to the COVID-19 pandemic, we had to conduct telephone interviews, during which some information might have been lost (e.g., context of interview situation, etc.). However, interviews could be scheduled more flexibly in terms of location and time, which might have facilitated the arrangement and implementation of interviews for the midwives.

We cannot rule out that participants with a special interest in ECAP and HL might have been more willing to participate in such a time-consuming interview study. Therefore, the interviewees were possibly better informed on ECAP or more aware of HL than the average German midwife and might have focused more on HL-sensitive counselling than others. Another limitation also concerns the sample: the educational level of the participants was fairly high, with 7 out of 24 midwives holding a university degree in subjects which go beyond “classic” midwifery, e.g., nursing science. This may indicate a strong interest in topics which lie beyond their daily work as midwives.

## 11. Conclusions

Midwives are health professionals who support families at a vulnerable and transitional time. As they are close to families, they may have an impact on the prevention of chronic diseases, like allergies, and preventive health behavior in general. They have the opportunity to enhance parents’ HL and thus to empower them to make informed choices on preventive behavior for their children. The midwives included in the sample of our qualitative study were mostly unaware of the concept of HL, formal screening strategies for parental HL, and HL-sensitive counselling techniques. This would suggest that further research on HL-sensitive counselling on ECAP on a larger scale is needed, in order to assess midwives’ awareness of the relevance of HL and their routine application of HL-sensitive counselling techniques in a broader, representative sample. Results of such future research could provide the basis for an intervention aimed at strengthening the HL-sensitive counselling capabilities of midwives in the prevention of allergies and other chronic diseases.

## Figures and Tables

**Figure 1 ijerph-19-04182-f001:**
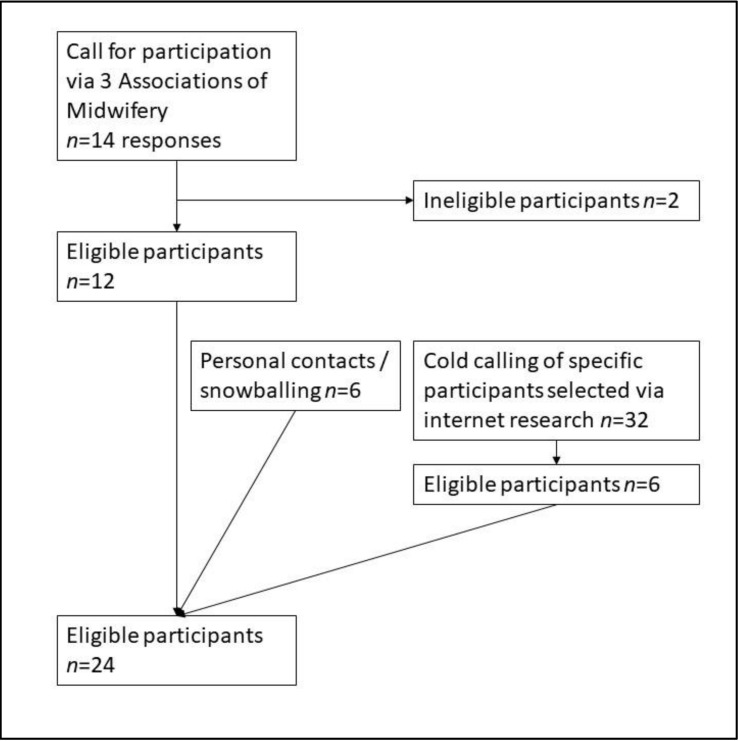
Flowchart of recruiting process.

**Table 1 ijerph-19-04182-t001:** Inclusion criteria of midwives participating in this study.

Inclusion Criteria	Description
Catchment area	Rural vs. urban
Education	Professional training vs. bachelor’s/master’s
Professional experience	15 years + vs. less than 15 years

**Table 2 ijerph-19-04182-t002:** Midwives included in our study.

	Midwives (*n* = 24)
Catchment area	
Village/small town (<20,000 inhabitants (IN))	14
Medium-sized/large town (>20,000 IN)	10
Education	
Vocational training	17
bachelor’s/master’s	7
Professional experience	
<10 years	12
>10 years	12

## Data Availability

The data presented in this study are available on request from the corresponding author. The data are not publicly available due to the need to preserve the anonymity of the interview partners.

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
