# Peer review of "Health Literacy-Sensitive Counselling on Early Childhood Allergy Prevention: Results of a Qualitative Study on German Midwives’ Perspectives"

_ijerph, 2022, doi:10.3390/ijerph19074182_

Round 1

Reviewer 1 Report

I congratulate the authors for a wonderful work done in this research. Using a qualitative approach often difficult to present with low bias or influence of thought by the authors. It is also very difficult to maintain thought flow through out such study without the readers loosing concentration. However, the author tried to avoid these challenges of qualitative research.

Author Response

Dear Reviewer 1,

thank you for your kind comments on our paper. We followed your suggestion regarding the spelling mistakes. An English native speaker reread our text and corrected some spelling and punctuation mistakes.

Sincerely,

the authors

Reviewer 2 Report

hi!
I wish you well in pandemic times. I apologize for the delay in delivering the paracer, but my demands are high.
The manuscript deals with a relevant and current subject. The research seems to me to have been well conducted. Therefore, I suggest small adjustments:
-In the abstract, please make clear the objective of the work.
-In the method, please insert a recruitment flowchart, as it helps to visualize the selection flow.
-In conclusion, the authors mention that "midwives in our sample were mostly unaware of the concept of HL, formal screening strategies for parental LH and sensitive LH
counseling techniques. This suggests that more research is needed on LS-sensitive ECAP counseling on a larger scale in order to assess midwives' knowledge of the
relevance of SL and its routine application of counseling techniques sensitive to HL in a broader and more representative sample".
Publication success.

Reviewer 3 Report

See attached

Author Response

Dear Reviewer 3,

thank you very much for your positive comments on our paper.

Sincerely,

the authors

Reviewer 4 Report

I would like to mention the following comments:

1- Keywords: It might be better to use "midwives" instead of "Health professionals".

2- Method: It might be better to also ask the opinions of "women specialists".

3- There is not enough explanation about the selection process of participants. Were they representative of all midwives?

4-  It might be better to have also opinions of pregnant women.

5- Some extra tables to summarize the various perspectives are needed.

Good Luck

Author Response

Please see attachement.

Reviewer 5 Report

Although the study might be the first qualitative study with the aim of understanding how German midwives engage in preventive counselling, how they take HL into account, and how they apply HL counselling techniques, there seems to be major concerns in the study.

1) Qualitative examination performed by the author cannot lead the conclusion showing that the midwives sample were mostly unware of the concept of HL, formal screening strategies for parental HL, and HL-sensitive counselling techniques. At least, using a little larger sample, the author should analyze at least the tendency.

2) How did the author select the 24 samples in the study? Sampling would have produced the selection bias.

3) The author should provide more detailed evidence for the effects of health literacy (H)L-sensitive counselling in early childhood allergy prevention (ECAP) on the subsequent outcome in children.

4) The author should describe clearly the difference between health literacy (H)L-sensitive counselling in early childhood allergy prevention (ECAP) and other intervention. Indeed, I understand the intervention of no smoking and regular exercise as the commune methods, it seems important to intervene the prevention of obesity or chronic cardiovascular diseases.

Round 2

Reviewer 5 Report

The manuscript has been well revised following the suggestion.